# Association between Neurologic Outcomes and Changes of Muscle Mass Measured by Brain Computed Tomography in Neurocritically Ill Patients

**DOI:** 10.3390/jcm11010090

**Published:** 2021-12-24

**Authors:** Yun Im Lee, Ryoung-Eun Ko, Joonghyun Ahn, Keumhee C. Carriere, Jeong-Am Ryu

**Affiliations:** 1Department of Internal Medicine, National Cancer Center, Goyang 10408, Korea; twirline@gmail.com; 2Department of Critical Care Medicine, Samsung Medical Center, Sungkyunkwan University School of Medicine, Seoul 06351, Korea; ryoungeun.ko@samsung.com; 3Statistic and Data Center, Clinical Research Institute, Samsung Medical Center, Seoul 06351, Korea; jhguy.ahn@samsung.com; 4Department of Mathematical and Statistical Sciences, University of Alberta, Edmonton, AB T6G 2G1, Canada; kccarrie@ualberta.ca; 5Department of Neurosurgery, Samsung Medical Center, Sungkyunkwan University School of Medicine, Seoul 06351, Korea

**Keywords:** neurosurgery, intensive care unit, sarcopenia, skeletal muscle mass, brain computed tomography

## Abstract

This study aimed to investigate whether skeletal muscle mass estimated via brain computed tomography (CT) could predict neurological outcomes in neurocritically ill patients. This is a retrospective, single-center study. Adult patients admitted to the neurosurgical intensive care unit (ICU) from January 2010 to September 2019 were eligible. Cross-sectional areas of paravertebral muscles at the first cervical vertebra level (C1-CSA) and temporalis muscle thickness (TMT) on brain CT were measured to evaluate skeletal muscle mass. The primary outcome was the Glasgow Outcome Scale score at 3 months. Among 189 patients, 81 (42.9%) patients had favorable neurologic outcomes. Initial and follow-up TMT values were higher in patients with favorable neurologic outcomes compared to those with poor outcomes (*p* = 0.003 and *p* = 0.001, respectively). The initial C1-CSA/body surface area was greater in patients with poor neurological outcomes than in those with favorable outcomes (*p* = 0.029). In multivariable analysis, changes of C1-CSA and TMT were significantly associated with poor neurological outcomes. The risk of poor neurologic outcome was especially proportional to changes of C1-CSA and TMT. The follow-up skeletal muscle mass measured via brain CT at the first week from ICU admission may help predict poor neurological outcomes in neurocritically ill patients.

## 1. Introduction

Nutrition is an important factor in the management of critically ill patients [1,2,3]. Malnutrition is associated with prolonged hospitalization and duration of mechanical ventilation, infection, and mortality in the intensive care unit (ICU) [2,4,5]. Malnutrition is also associated with poor clinical outcomes in neurocritically ill patients [6,7,8]. Nutritional support can affect neurological prognosis as well as mortality in patients with stroke or traumatic brain injury [6,7,8]. Sarcopenia is characterized by the loss of skeletal muscle mass and its function [9]. Skeletal muscle mass is associated with physiologic functions [2,10,11]. In critically ill patients, malnutrition and prolonged immobility due to severe illness increase the risk of sarcopenia during their ICU stay [2]. Eventually, sarcopenia is associated with poor clinical prognosis in severely ill patients [12,13,14]. Therefore, it is important to estimate the nutritional status based on skeletal muscle mass and to provide adequate nutrition.

Skeletal muscle mass can be measured via whole body or regional dual-energy X-ray absorptiometry scans and volumetric or cross-sectional area (CSA) measurements on magnetic resonance imaging or computed tomography (CT) scans at the arm, leg, or third lumbar vertebral level [13,15]. However, muscle mass measurement using the CSA on imaging scans and dual-energy X-ray absorptiometry scans may not be routinely performed in neurocritically ill patients [16]. In neurocritically ill patients, brain CT scans are frequently performed. Although skeletal muscle mass is not routinely assessed in brain CT, it may rapidly decrease on follow-up brain CT scans in neurocrtically ill patients (Figure 1). A small number of studies evaluated skeletal muscle mass via brain CT [16,17,18]. In addition, few studies reported clinical prognosis according to the changes in skeletal muscle mass using brain CT. Therefore, the objective of this study was to investigate whether skeletal muscle mass estimated via brain CT was associated with neurological outcomes of neurocritically ill patients.

## 2. Materials and Methods

### 2.1. Study Population

This is a retrospective, single-center, observational study. Adult patients who were admitted to the neurosurgical ICU in our tertiary hospital (Samsung Medical Center, Seoul, Republic of Korea) from January 2010 to September 2019 were eligible. This study was approved by the Institutional Review Board of Samsung Medical Center (approval number: SMC 2020-02-113). The requirement for informed consent was waived due to its retrospective nature. We included patients (1) who were hospitalized in the neurosurgical ICU for more than 7 days, (2) were evaluated with brain CT on ICU admission, and (3) had a follow-up brain CT within the first 6 to 8 days after ICU admission. Of these patients, we excluded patients (1) aged below 18 years, (2) who did not have a brain injury, (3) who had insufficient medical records, (4) who have a history of chronic neurological abnormality on admission, (5) who stayed in the ICU for more than 7 days due to the lack of a general ward, (6) who were on a ‘do not resuscitation’ order, and (7) who were admitted to departments other than neurosurgery (Figure 2).

### 2.2. Definitions and Endpoints

In this study, baseline characteristics of comorbidities, causes of ICU admission, and initial clinical parameters on admission were retrospectively obtained through medical record review. Acute Physiology and Chronic Health Evaluation (APACHE) II score was calculated with the worst values recorded during the initial 24 h after the ICU admission [19,20]. If the patient was intubated, the verbal score of the Glasgow Coma Scale was estimated using the eye and motor scores as described previously [21].

CSA of the paravertebral muscle at the first cervical vertebral level (C1-CSA) was evaluated on brain CT (Figure 3A). The imaging protocol of brain CT performed in our hospital always includes the first cervical vertebra, but lower vertebral levels are sometimes not included in the scan images. Therefore, the first cervical vertebra was chosen as a point of measurement. The skeletal muscles were identified at the transverse process level with Hounsfield unit thresholds ranging from −29 to +150. An investigator delineated all the muscles manually, and the C1-CSA was automatically retrieved as the total sum of delineated pixels [16]. The difference between the initial C1-CSA and follow-up C1-CSA (∆C1-CSA) was defined as the initial C1-CSA minus the follow-up C1-CSA. The change of C1-CSA was defined as ∆C1-CSA divided by the initial C1-CSA multiplied by 100. Temporalis muscle thickness (TMT) was also measured perpendicularly to the long axis of the temporal muscle in the axial plane of the CT image (Figure 3B). The Sylvian fissure was used as a reference point of TMT measurement at the level of the orbital roof [18,22]. The maximum TMT was used as the TMT value, whichever was thicker than the other. If the patient underwent neurosurgery, including craniotomy or craniectomy, on one side within two weeks before the initial brain CT scan, the TMT of the other side alone was used for analysis. If the patient had neurosurgery bilaterally, their TMT values were not used in the analysis. The difference between initial TMT and follow-up TMT (∆TMT) was defined as the initial TMT minus the follow-up TMT. The change of TMT was defined as ∆TMT divided by the initial TMT multiplied by 100. All the CT studies were performed using 64-channel scanners (Light Speed VCT, GE Healthcare, Milwaukee, WI, USA) with a 5-mm slice width. Trained intensivists evaluated each of the patients’ CT scans using commercial image-viewing software (Centricity RA1000 PACS Viewer, GE Healthcare, Milwaukee, WI, USA) [23]. The images were changed to the “chest/abdomen” window (window width 300 & window level 10) and magnified threefold to fourfold on the particular image slice that demonstrated the largest diameter of TMT.

The primary outcome was the performance on the Glasgow Outcome Scale (GOS) at 3 months. Patients with GOS scores of 4 to 5 indicated favorable neurological outcomes, whereas GOS scores of 1 to 3 suggested poor neurological outcomes [24,25].

### 2.3. Statistical Analyses

All data are presented as medians and interquartile ranges (IQRs, Q1~Q3) for continuous variables and as numbers (percentages) for categorical variables. Data were compared using the Mann-Whitney *U* test for continuous variables and the Chi-square test or Fisher’s exact test for categorical variables. Variables with a *p*-value less than 0.2 in univariable analyses and clinically relevant variables, including age, sex, BMI, BSA, comorbidities, GCS and APACHE II score on ICU admission, the initial level of serum albumin, and use of mannitol, were subjected to multiple logistic regression analysis to obtain statistically meaningful predictors. Stepwise variable selection was conducted to construct the final model. Adequacy of the prediction model was determined using the Hosmer-Lemeshow test, along with the areas under the curve (AUC). Split-sample analyses and 10-fold cross-validation analyses were conducted to assess the internal validity. All tests were two-sided and *p* < 0.05 was considered statistically significant. All data were analyzed using R Statistical Software (version 4.0.2; R Foundation for Statistical Computing, Vienna, Austria).

## 3. Results

### 3.1. Baseline Characteristics and Clinical Outcomes

A total of 189 patients were analyzed (Figure 2). The median age of patients was 58.0 (IQR: 48.0–70.0) years. One hundred patients (52.9%) were males. Brain tumor (41.3%) and stroke (37.0%) were the most common causes of ICU admission. Malignancy (56.1%) and hypertension (46.6%) were the most common comorbidities in the study population. Age and APACHE II scores on ICU admission were greater in the poor outcome group than in the favorable outcome group. Body mass index (BMI) and body surface area (BSA) were higher in the favorable outcome group compared with the poor outcome group. Baseline characteristics of the study population are presented in Table 1.

Among the 189 patients, 167 (88.4%) survived until discharge from the hospital. Of these survivors, 81 patients had favorable neurologic outcomes. The entire distribution of GOS is shown in Figure 2.

### 3.2. Relationship between C1-CSAs, TMTs, and Neurological Outcomes

Initial and follow-up TMT values were higher in patients with favorable neurological outcomes compared to those with poor neurological outcomes. However, the changes of TMT and ∆TMT were not significantly different between the two groups. Although initial C1-CSA/BSA was greater in patients with the poor neurological outcome than in favorable outcomes (*p* = 0.029), other variables related to CSA were not significantly different between the two groups (Table 2).

The process of selecting variables for the regression model is presented in Figure 4. The results of the simple and multiple logistic regression model are presented in Table 3 and Figure 5. In multivariable analysis (Table 3), age (adjusted odds ratio {OR}: 2.05, 95% confidence interval {CI}: 1.543–2.724), BMI (adjusted OR: 0.74, 95% CI: 0.638–0.849), use of mannitol (adjusted OR: 27.45, 95% CI: 4.833–155.860), change of C1-CSA (adjusted OR: 1.36, 95% CI: 1.054–1.761), and change of TMT (adjusted OR: 1.27, 95% CI: 1.028–1.576) were significantly associated with poor neurological outcome (Hosmer–Lemeshow Chi-squared = 11.4, *df* = 8, *p* = 0.178) with the AUCs of 0.803 (95% CI 0.740–0.866) using a 10-fold cross-validation method (Figure 6). The risk of poor neurological outcome was especially proportional to changes of C1-CSA and TMT (Figure 7).

In the final model, the change of C1-CSA and that of TMT were compared to see whether one factor was superior in predicting the clinical outcome than the other was. Various statistical methods such as the standardized regression coefficients and c-stats were used. In conclusion, the change of TMT and that of C1-CSA affected the clinical outcome in a similar way. Moreover, the relationship between these two factors was analyzed and multicollinearity was not found between two factors.

## 4. Discussion

In this study, we investigated whether the skeletal muscle mass estimated by brain CT could be used to predict neurological outcomes in neurocritically ill patients. The major findings of this study were as follows: First, half of the surviving patients had a favorable neurological prognosis in this study. Second, during the initial and follow-up CT, the TMT values of the poor neurological outcome group were significantly lower than those of the favorable neurological outcome group. However, during the initial and follow-up CT, the C1-CSAs were not significantly different between the two groups except for the initial C1-CSA/BSA. Third, in multivariable analysis, age, BMI, use of mannitol, and changes in C1-CSA and TMT were significantly associated with poor neurological outcomes in neurocritically ill patients. The risk of poor neurological outcome was especially proportional to changes of C1-CSA and TMT.

Nutritional support is an important issue in the intensive care of critically ill patients [1,2,3]. Malnutrition is associated with poor clinical prognosis of neurocritically ill patients [6,7,26]. Inadequate nutritional support increases susceptibility to infection, mortality, and neurological outcomes in these patients [6,7,26,27]. Malnutrition has been estimated depending on various parameters that may include BMI, serum albumin, and skeletal muscle mass [2]. However, BMI and serum albumin are poor parameters representing nutritional status in critically ill patients [1,2]. Skeletal muscle mass is a more accurate parameter in assessing nutritional status and may reflect the clinical prognosis better than other nutritional measures in critically ill patients [2].

The CSA of skeletal muscle mass has been estimated via abdominal CT at the third lumbar vertebral level, which correlates with the total body skeletal muscle mass and can be easily measured on an abdominal CT acquired during intensive care [12,16,28,29]. Recent studies showed that CSAs of skeletal muscle mass at the level of cervical vertebrae on a head and neck CT scan significantly correlate with those at the third lumbar vertebral level on abdominal CT scan [16,30]. In addition, TMT also correlates with CSAs of skeletal muscle mass at the third lumbar vertebral level or total psoas muscle area on abdominal CT scans [17,18]. Therefore, CSAs of skeletal muscle mass at the cervical vertebra levels and TMT on brain CT can be used as alternatives to estimate sarcopenia and nutritional status in neurocritically ill patients.

Cervical skeletal muscles and facet joints are crucial parts in facilitating the movement of the cervical spine. Along with intervertebral discs and spinal ligaments, the cervical muscle and facet joints bind the adjacent vertebrae and bear loads applied to the cervical spinal column [31]. When the cervical spine is injured, clinicians usually use CT or magnetic resonance imaging to evaluate the extent of injury [32]. In the previous publications, Ulbrich et al. measured the CSA of cervical muscles in patients who experienced whiplash injury using magnetic resonance imaging. In those studies, they found that CSA values of cervical muscle were associated with BMI [33,34]. Though BMI is not an exact reflection of muscle mass, CSA of cervical muscle can be used to assess patients’ nutritional status. Sarcopenia generally occurs in critically ill patients and may progress after ICU admission [35]. Skeletal muscle mass begins to decrease remarkably within 3 days and gradually deteriorates [3,35]. In addition, the muscle mass of the limbs can be reduced by one-fifth within 7 days after ICU admission due to malnutrition and prolonged immobility as a consequence of critical illness [35,36]. Skeletal muscle mass plays an important role in physiological functions such as immune modulation, protein synthesis, and glucose metabolism [2,10,11]. Therefore, sarcopenia secondary to critical illness is associated with adverse clinical prognosis [12,13]. Similarly, malnutrition during the first week could be associated with poor neurological outcomes in patients with stroke [6]. Therefore, sarcopenia in the first week may be associated with poor neurological outcomes in neurocritically ill patients as well. In this study, changed muscle mass at the first week was also associated with prognosis in neurocritically ill patients.

This study has several limitations. First, this was a retrospective review. Thus, GOS was determined based on medical records. Any bias involving the scores was mitigated partially based on the consensus of two independent specialists. Second, the nonrandomized nature of registry data might have resulted in selection bias. Brain CT scans were not protocol-based in their performance. Third, TMT of the surgical direction was not available because of possible damage and mobilization of the temporalis muscle occurring during either dissection, transection, or incision after temporal craniotomy [37]. Fourth, the median ages were significantly different between study groups and that could have affected the clinical outcomes. Fifth, in the case of patients with cervical facet injury, CSA of cervical muscle may not be measured correctly with CT [32]. Lastly, our study has limited statistical power due to its small sample size. Although it still provides valuable insight, prospective large-scale studies are needed to confirm the role of brain CT-based muscle mass measurement in predicting the clinical prognosis of neurocritically ill patients to arrive at evidence-based conclusions.

## 5. Conclusions

In this study, follow-up skeletal muscle mass at the first week from ICU admission based on changes in C1-CSA and TMT is associated with neurological prognosis in neurocritically ill patients. Eventually, sarcopenia measured via brain CT may suggest poor neurological outcomes in these patients. Therefore, adequate nutritional support and early mobilization to prevent sarcopenia may facilitate recovery in neurocritically ill patients.

## Figures and Tables

**Figure 1 jcm-11-00090-f001:**
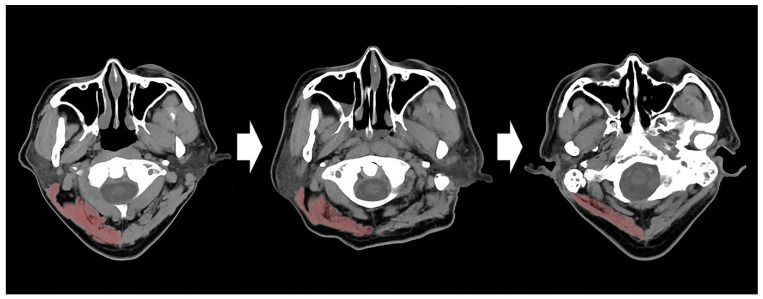
Serial changes of cervical muscle mass in a patient with traumatic brain injury. The brain CT images with 7 days intervals showed a progressive decrease of the cervical muscle mass. CT, computed tomography.

**Figure 2 jcm-11-00090-f002:**
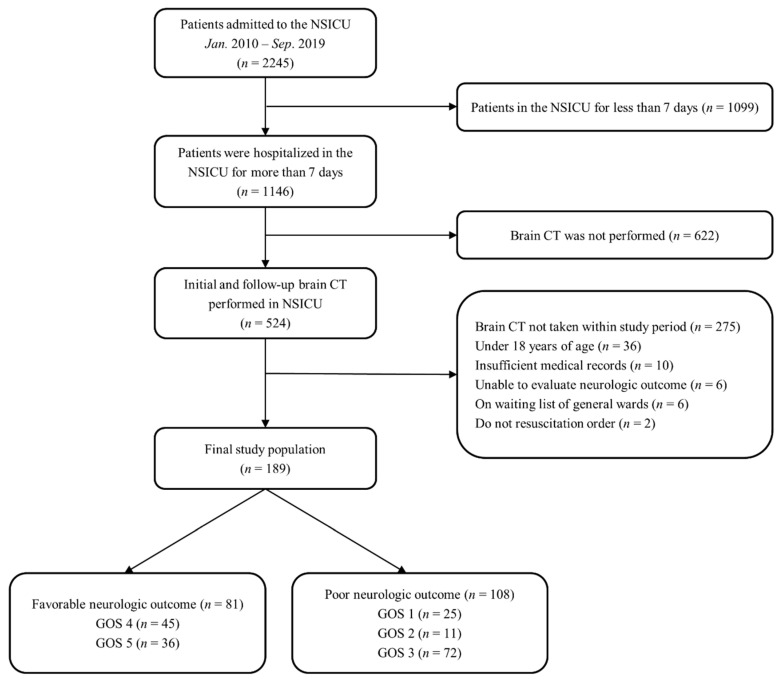
Study flow chart. NSICU, neurosurgical intensive care unit; CT, computed tomography; and GOS, Glasgow Outcome scale.

**Figure 3 jcm-11-00090-f003:**
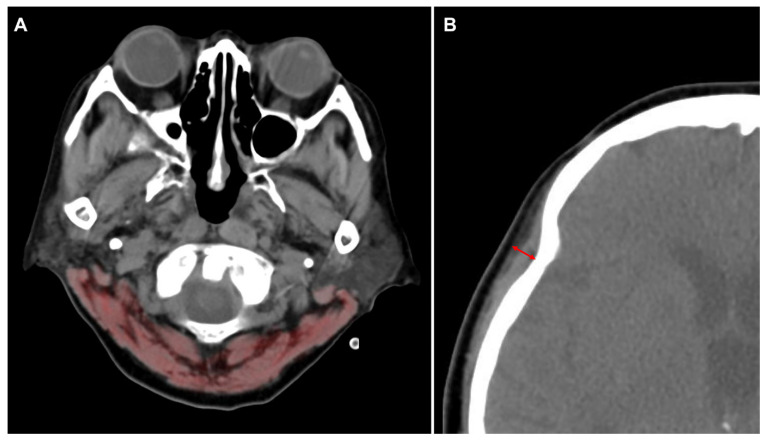
Methods for the measurement of the cross-sectional area at the level of first cervical vertebra (C1-CSA) (**A**) and temporalis muscle thickness (TMT) (**B**) on brain computed tomography images.

**Figure 4 jcm-11-00090-f004:**
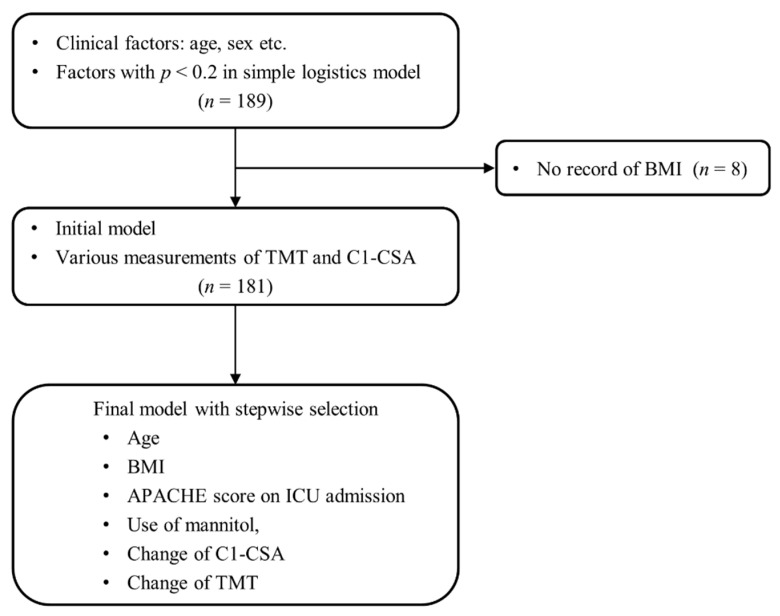
The process of selecting variables for the regression model. Abbreviations: BMI, body mass index; ICU, intensive care unit; APACHE, Acute Physiology and Chronic Health Evaluation; C1-CSA, the cross-sectional area of the paravertebral muscle at first cervical vertebra; and TMT, temporalis muscle thickness.

**Figure 5 jcm-11-00090-f005:**
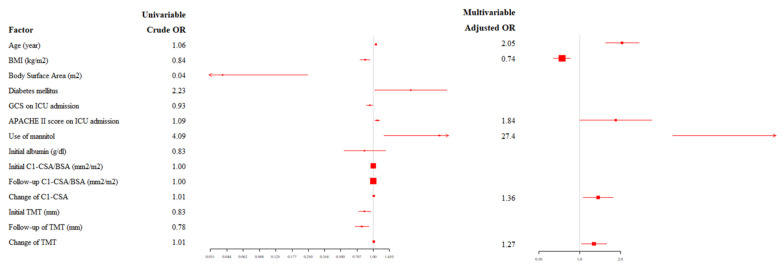
Forest plot of the regression model. Abbreviations: BMI, body mass index; GCS, Glasgow Coma Scale; ICU, intensive care unit; APACHE, Acute Physiology and Chronic Health Evaluation; BSA, body surface area; Change of C1-CSA, 100 × (initial C1-CSA minus follow-up C1-CSA/initial C1-CSA); and change of TMT, 100 × (initial TMT minus follow-up TMT)/initial TMT.

**Figure 6 jcm-11-00090-f006:**
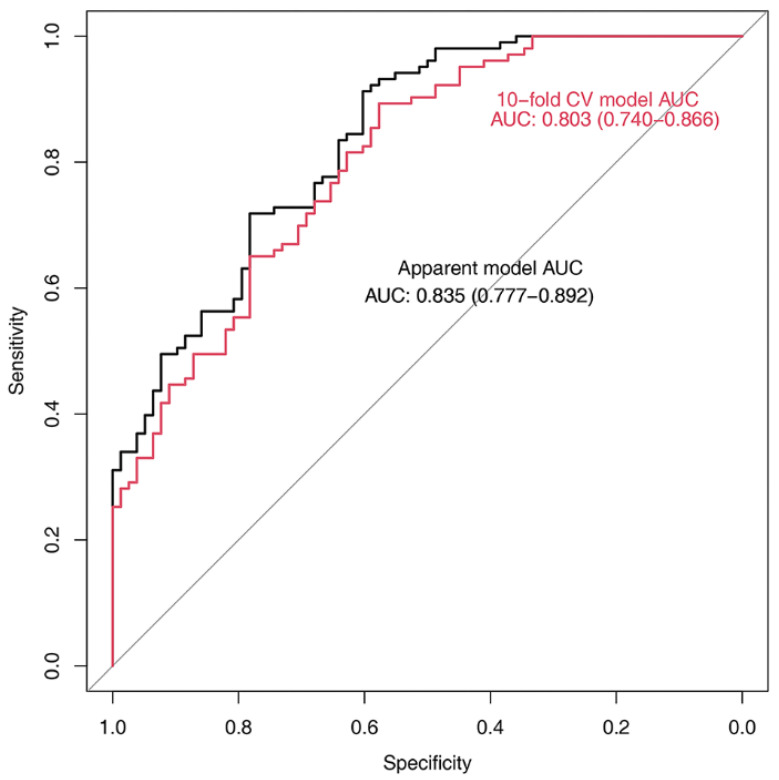
Adequacy of the prediction model for poor neurological outcomes in this study as determined using the Hosmer-Lemeshow test (Chi-squared = 11.4, *df* = 8, *p* = 0.178), along with the areas under the curve (AUC: 0.803, 95% confidence interval: 0.740–0.866). In this study, a 10-fold cross-validation (CV) analysis was conducted to assess the internal validity.

**Figure 7 jcm-11-00090-f007:**
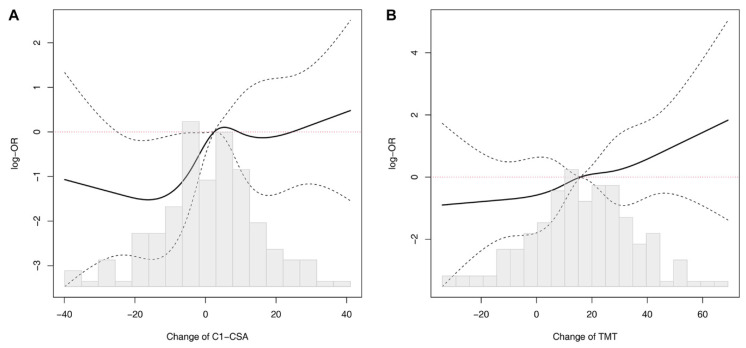
Association between changes in C1-CSA (**A**) or TMT (**B**) and poor neurologic outcome. OR, odds ratio; C1-CSA, the cross-sectional area at the level of the first cervical vertebra; TMT, temporalis muscle thickness; Change of C1-CSA, 100 × (initial C1-CSA minus follow-up C1-CSA)/initial C1-CSA; and change of TMT, 100 × (initial TMT minus follow-up TMT)/initial TMT.

**Table 1 jcm-11-00090-t001:** Baseline characteristics.

	Favorable Neurologic Outcome (*n* = 81)	Poor Neurologic Outcome (*n* = 108)	*p* Value
Age (year)	53.0 (33–63.5)	63.5 (52.3–72.8)	<0.001
Sex, male	41 (50.6)	59 (54.6)	0.689
BMI (kg/m^2^)	24.1 (22.6–26.7)	22.8 (20.7–25.1)	<0.001
Body surface area (m^2^)	1.8 (1.6–1.8)	1.6 (1.5–1.8))	<0.001
Comorbidities			
Malignancy	46 (56.8)	60 (55.6)	0.983
Hypertension	30 (37.0)	55 (50.9)	0.080
Diabetes mellitus	11 (13.6)	28 (25.9)	0.058
Current smoker	13 (16.0)	15 (13.9)	0.836
Ischemic heart disease	4(4.9)	8 (7.4)	0.698
Chronic kidney disease	4 (4.9)	8 (7.4)	0.698
Cause of ICU admission			0.027
Brain tumor	34 (42.0)	44 (40.7)	
Stroke *	29 (35.8)	41 (38.0)	
Traumatic brain injury	4 (4.9)	16 (14.8)	
Others	14 (17.3)	7 (6.5)	
GCS on ICU admission	7.0 (3.0–13.0)	6.0 (3.0–10.0)	0.030
APACHE II score on ICU admission	18.0 (14.0–23.0)	21.0 (17.3–26.0)	0.001
Use of mannitol ^†^	70 (86.4)	104 (96.3)	0.027
Use of glycerin ^†^	54 (66.7)	75 (69.4)	0.804
Use of dexamethasone	57 (70.4)	68 (63.0)	0.363
Initial albumin level (g/dL)	3.4 (3.1–3.9)	3.4 (3.0–3.9)	0.403

Data are numbers (%) or median (interquartile range). * Stroke included intracerebral hemorrhage, subarachnoid hemorrhage, and cerebral infarction. ^†^ Some patients received more than one hyperosmolar agent. Abbreviations: BMI, body mass index; ICU, intensive care unit; GCS, Glasgow Coma Scale; and APACHE, Acute Physiology and Chronic Health Evaluation.

**Table 2 jcm-11-00090-t002:** The cross-sectional areas (CSAs) of first cervical vertebra level (C1) and temporalis muscle thicknesses (TMTs) according to neurological outcomes.

	Favorable Neurologic Outcome (*n* = 81)	Poor Neurologic Outcome (*n* = 108)	*p* Value
Initial C1-CSA (mm^2^)	1825.2 (1602.4–2165.3)	1853.9 (1605.1–2206.6)	0.495
Initial C1-CSA/BSA (mm^2^/m^2^)	1071.5 (952.0–1225.4)	1120.4 (1040.4–1299.0)	0.029
Follow-up C1-CSA (mm^2^)	1850.0 (1598.3–2150.6)	1807.8 (1577.1–2089.1)	0.686
Follow-up C1-CSA/BSA (mm^2^/m^2^)	1072.6 (930.6–1201.8)	1099.4 (978.5–1231.8)	0.390
∆C1-CSA (mm^2^)	22.8 (−147.3–180.6)	78.1 (−86.3–225.7)	0.123
∆C1-CSA/BSA (mm^2^/m^2^)	7.5 (−84.8–111.3)	60.0 (−42.4–137.7)	0.086
Change of C1-CSA	1.4 (−7.9–9.4)	4.4 (−4.4–11.6)	0.133
Initial TMT (mm)	7.2 (6.1–9.1)	6.4 (5.2–7.6)	0.003
Follow-up TMT (mm)	5.9 (4.9–7.6)	5.1 (4.0–6.6)	0.001
∆TMT (mm)	1.0 (0.2–1.9)	1.3 (0.4–2.1)	0.496
Change of TMT	14.1 (−2.9–26.5)	18.1 (7.9–29.6)	0.110

Data are median (interquartile range). Abbreviations: BSA, Body surface area; ∆C1-CSA, initial C1-CSA minus follow-up C1-CSA; ∆C1-CSA/BSA, initial C1-CSA/BSA minus follow-up C1-CSA/BSA; Change of C1-CSA, 100 × (∆C1-CSA/initial C1-CSA); ∆TMT, initial TMT minus follow-up TMT; and change of TMT, 100 × (initial TMT minus follow-up TMT)/initial TMT.

**Table 3 jcm-11-00090-t003:** Association between factors and poor neurologic outcome at 3 months.

	Univariable Analysis		Multivariable Analysis	
	Crude Odds Ratio (95% CI)	*p* Value	Adjusted Odds Ratio (95% CI)	*p* Value
Age (year)	1.06 (1.033–1.078)	<0.001	2.05 (1.543–2.724)	<0.001
BMI (kg/m^2^)	0.84 (0.757–0.930)	0.001	0.74 (0.638–0.849)	<0.001
Body surface area (m^2^)	0.04 (0.007–0.247)	<0.001		
Diabetes mellitus	2.23 (1.034–4.799)	0.041		
GCS on ICU admission	0.93 (0.865–0.991)	0.026		
APACHE II score on ICU admission	1.09 (1.035–1.143)	0.001	1.84 (0.996–3.396)	0.052
Use of mannitol	4.09 (1.251–13.347)	0.020	27.4 (4.833–155.860)	<0.001
Initial albumin level (g/dL)	0.83 (0.535–1.298)	0.420		
Initial C1-CSA/BSA (mm^2^/m^2^)	1.00 (1.000–1.003)	0.026		
Follow-up C1-CSA/BSA (mm^2^/m^2^)	1.00 (1.000–1.002)	0.155		
Change of C1-CSA	1.01 (0.989–1.026)	0.432	1.36 (1.054–1.761)	0.018
Initial TMT (mm)	0.83 (0.733–0.945)	0.005		
Follow-up TMT (mm)	0.78 (0.677–0.909)	0.001		
Change of TMT	1.01 (0.993–1.019)	0.383	1.27 (1.028–1.576)	0.027

Abbreviations: BMI, body mass index; GCS, Glasgow Coma Scale; ICU, intensive care unit; APACHE, Acute Physiology and Chronic Health Evaluation; BSA, body surface area; Change of C1-CSA, 100 × (initial C1-CSA minus follow-up C1-CSA/initial C1-CSA); and Change of TMT, 100 × (initial TMT minus follow-up TMT)/initial TMT.

## Data Availability

Regarding data availability, our data are available on the Harvard Dataverse Network (http://dx.doi.org/10.7910/DVN/GF08RY, accessed on 1 December 2020).

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
