# Peer review of "Association between Neurologic Outcomes and Changes of Muscle Mass Measured by Brain Computed Tomography in Neurocritically Ill Patients"

_jcm, 2021, doi:10.3390/jcm11010090_

Round 1

Reviewer 1 Report

The study by Dr. Lee and colleagues highlight an important issue of prognosis for critically ill neurological/neurosurcial patients who often require prolonged treatment in the ICU setting. Authors demonstrate how to use readily available data for prognostication.

Regarding data analysis, more detailed description of testing procedure is suggested, as it may be subject to bias; running univariate comparisons first in order to select variables to use in the final multiple regression model might be suboptimal due to endogeneity problem. I would also suggest that Table 3 should include full results of the testing (i.e., tested parameters, that did not reach significance, besides APACHE II score). These results may be also more illustrative if presented as a diagram.

In general, the results may pave the way for further development of the method.

Suggestions for some minor changes:

Lines 139-140: primary diseases (brain tumor and stroke) should be desribed before comorbidities (malignancy and hypertension).

Line 203: "Third" instead of "Second".

Reviewer 2 Report

The authors focused on a study of the Prognostic Value of Muscle Mass Measured via Brain Computed Tomography in Neurocritically Ill Patients. This is an interesting but some parts are missing some important facts that authors should add. The article is well structured. Three tables and five figures in the text are very clearly written.

In my opinion:

  • Title is not clearly described in the article.
  • The Abstract does not present an accurate description of the case and its implications - a retrospective review !!!
  • An Authors was conducted adequate literature review.
  • The references support the rationale for reporting the study.
  • The patients are described adequately.
  • The management of the study is effectively described.
  • Valid and reliable outcome measures are utilized.
  • The conclusions are appropriate.
  • The prospective large-scale studies are needed to confirm your study.

Overall impression about the quality of the study is good.

Reviewer 3 Report

Interesting, but some points require revision:

  • Lines 57-58: "A limited number of studies evaluated the skeletal muscle mass via brain CT [13-15]" Although the meaning that authors want to give to this sentence is understood, it must nevertheless be revised.
  • Lines 58-59: "no study reported clinical prognosis according to the changes in skeletal muscle" Look at these refs: Measuring and monitoring skeletal muscle function in COPD: current perspectives. Int J Chron Obstruct Pulmon Dis. 2019 Aug 19;14:1825-1838. -- Skeletal muscle loss and prognosis of breast cancer patients. Support Care Cancer. 2017 Jul;25(7):2221-2227.
  • From the statistical analysis, it seems that the thicknesses of the temporal muscles appear to be more prognostic than cervical muscle mass. This should be discuss more in the text.
  • Lines 223-224: "Therefore, CSAs of skeletal muscle mass at the cervical vertebra levels and TMT on brain CT can be used as alternatives to estimate sarcopenia and nutritional status in neurocritically ill patients" at this point, authors should discuss more about the role of the cervical muscle mass and the facet joints. Consider these refs:  -- Regional and experiential differences in surgeon preference for the treatment of cervical facet injuries: a case study survey with the AO Spine Cervical Classification Validation Group. Eur Spine J. 2021 Feb;30(2):517-523. doi: 10.1007/s00586-020-06535-z.   --  Sarcopenia and Muscle Aging: A Brief Overview. Endocrinol Metab (Seoul). 2020 Dec;35(4):716-732. 
  • Age of the 2 groups is different and this must be reported between limitations of the paper.
